# Deep Visual Computing of Behavioral Characteristics in Complex Scenarios and Embedded Object Recognition Applications

**DOI:** 10.3390/s24144582

**Published:** 2024-07-15

**Authors:** Libo Zong, Jiandong Fang

**Affiliations:** 1College of Information Engineering, Inner Mongolia University of Technology, Hohhot 010080, China; wxyzlibo@163.com; 2Inner Mongolia Key Laboratory of Perceptive Technology and Intelligent Systems, Hohhot 010080, China; 3Inner Mongolia Synergy Innovation Center of Perception Technology in Intelligent Agriculture and Animal Husbandry, Hohhot 010080, China

**Keywords:** classroom video, face recognition, identity authentication, object detection, teaching aids

## Abstract

By leveraging artificial intelligence and big data to analyze and assess classroom conditions, we can significantly enhance teaching quality. Nevertheless, numerous existing studies primarily concentrate on evaluating classroom conditions for student groups, often neglecting the need for personalized instructional support for individual students. To address this gap and provide a more focused analysis of individual students in the classroom environment, we implemented an embedded application design using face recognition technology and target detection algorithms. The Insightface face recognition algorithm was employed to identify students by constructing a classroom face dataset and training it; simultaneously, classroom behavioral data were collected and trained, utilizing the YOLOv5 algorithm to detect students’ body regions and correlate them with their facial regions to identify students accurately. Subsequently, these modeling algorithms were deployed onto an embedded device, the Atlas 200 DK, for application development, enabling the recording of both overall classroom conditions and individual student behaviors. Test results show that the detection precision for various types of behaviors is above 0.67. The average false detection rate for face recognition is 41.5%. The developed embedded application can reliably detect student behavior in a classroom setting, identify students, and capture image sequences of body regions associated with negative behavior for better management. These data empower teachers to gain a deeper understanding of their students, which is crucial for enhancing teaching quality and addressing the individual needs of students.

## 1. Introduction

With the rapid development and widespread application of information technology, the field of education has gradually embraced digital and intelligent transformations. Accelerating the digital transformation of education is not only an inevitable trend of the times but also a crucial means to promote the modernization of education and enhance its quality. Consequently, many researchers have started exploring advanced intelligent technologies to improve the teaching process, with deep learning-based student classroom analysis methods gaining significant attention. Students’ behaviors in the classroom directly reflect the quality of the educational experience. Yang et al. [1] conducted gesture recognition on key points of the human body detected in classroom images, effectively identifying various classroom behaviors. Huang et al. [2] employed a target detection model and a target tracking algorithm combined with a behavior recognition algorithm to identify students’ behaviors in the classroom. Additionally, capturing students’ emotions in the classroom is an effective means of accurately assessing the learning process. Shou et al. [3] utilizes facial expression and head pose data obtained from deep learning models to comprehensively analyze students’ academic emotions. Similarly, Wang et al. [4] assessed students’ concentration levels in the classroom based on their facial expressions and postures. It is evident that smart technologies have shown great potential in student classroom analysis and the digital development of education in China has made considerable progress. However, a key issue in this process is the over-emphasis on theoretical constructs and facility construction while neglecting practical implementation and application [5]. In the development of smart education, many studies and projects focus more on establishing theoretical models and building infrastructure rather than applying these theories and facilities to educational practice. This tendency creates a disconnect between theory and practice, hindering the full potential of intelligent technologies in practical teaching. Therefore, it is increasingly important to promote the research and development of digital education applications and leverage diverse intelligent technologies to assess academic learning conditions, aiming to further personalize and accurately guide the learning process. Combined with face recognition technology for automatic student identification, this approach shows significant potential in accurately assessing individual learning situations.

Deep convolutional neural networks have led to significant breakthroughs in face recognition technology, with many notable algorithms emerging in recent years, such as Cosface [6], Arcface [7], MagFace [8], and Adaface [9]. GaussianFace, proposed by Lu et al. [10] in 2015, achieved an accuracy of 98.52% on the LFW dataset, surpassing the human-level performance of 97.53% for the first time. All of the aforementioned algorithms have since exceeded this benchmark. To enhance the accuracy of student attendance recognition, Dang et al. [11] combined fingerprint scanning and face recognition for attendance management. Their face recognition model was based on the MobileNetV2 backbone network and SSD sub-module and it was deployed on the Jetson Nano. Additionally, Feng et al. [12] optimized the MobileNetV3 feature extraction network and employed both the Softmax and Center loss functions to jointly supervise the training, achieving face recognition in classroom scenarios. Tian et al. [13] employed the Mobilefacenet network structure within Insightface, optimized the parameter count, and integrated it into the Jetson TX2 development board for applications in classroom scenarios involving 8 to 18 students. The face recognition technologies in these studies were typically designed for scenarios with small numbers of students and did not sufficiently address applications involving posture changes and complex environments. Additionally, these studies focused on student attendance management rather than utilizing face recognition technology to record and track student behavior in the classroom.

In summary, research on classroom face recognition using deep convolutional neural networks has garnered significant attention. Building on classroom face recognition, research combining classroom analysis to provide personalized and accurate guidance for individual students still requires further exploration and development. To better focus on individual students in the classroom environment and facilitate personalized instruction, this paper employs Insightface face recognition technology to identify students and combines target detection algorithms to correlate body regions with their identities. The classroom face dataset is expanded through data collection and enhancement methods. The YOLO (You Only Look Once) algorithm is employed to detect student body regions and locate specific students by calculating containment relationships with facial regions. Finally, the model algorithms are deployed on an embedded device, the Atlas 200 DK (Developer Kit), enabling the interception and upload of classroom images associated with individual students to a server for management, thus providing a basis for classroom analytics and personalized instruction. The main contributions of this paper are as follows. Although there are many studies on students’ classroom analysis, there are few publicly available datasets. To address this, we have collected a classroom face dataset and a classroom behavior dataset. In classrooms with a large number of densely arranged students, identity identification can be easily confused due to the influence of angles and overlap. To mitigate this, we propose an identity identification algorithm by analyzing the relationship between the face and body. Based on the above, a system application has been developed for embedded devices, combining face recognition and target detection. This application provides a foundation for classroom analysis and personalized teaching.

## 2. Algorithm Principles

The structure of the embedded application system for student identification based on face recognition is shown in Figure 1, which contains four main modules: a classroom face detection module, a classroom face recognition module, a classroom student target detection module, and an identity identification and upload module. The classroom student face detection module extracts features from the classroom image via a deep convolutional neural network and subsequently applies a classification network to determine the confidence level of the student’s face and calculates its exact coordinates using a border regression network. The classroom student face recognition module captures student face images based on the results of face detection and then extracts key features from these images. Subsequently, the module is able to verify the identity of the student by comparing it with the pre-stored features in the database. The classroom student target detection module also utilizes deep convolutional neural networks to extract features from the video, which can be processed to accurately locate the student’s position. The identity identification and upload module is responsible for integrating the results of face recognition and target detection and finally intercepting and uploading the relevant images. Identity determination is crucial in this system; hence, selecting a reliable face recognition model is a priority. Since the face recognition model requires substantial resources, a relatively lightweight target detection model was chosen. Additionally, the solution must be easily embedded into the Atlas 200 DK, so the algorithms InsightFace and YOLOv5 were selected to work together. The system algorithm flowchart is shown in Figure 2, in which Insightface is responsible for face detection and feature extraction and YOLO is responsible for the detection of students’ body regions.

### 2.1. Face Recognition Process

The core steps of the face recognition algorithm cover face detection, image preprocessing, feature extraction, and final recognition. Among them, face image preprocessing, as a crucial link, provides a more accurate and standardized database for feature extraction and the recognition of face images by performing operations such as alignment and cropping, thus ensuring the accuracy and efficiency of the subsequent work.

#### 2.1.1. Face Detection

The one-stage Retinaface-based [14] algorithm for face detection was used in the Insightface project. The algorithm was trained on the WebFace 600K dataset using ResNet50 as the backbone feature extraction network. The feature maps extracted by the backbone network are used as the output. In order to enhance the feature extraction capability, the algorithm employs the feature pyramid technique, which realizes multi-scale feature fusion through Upsample and Add operations, thus enhancing the feature extraction effect. In addition, to further enhance the detection of small-sized faces, the algorithm introduces a context module to expand the sensory field, which improves the detection rate of small-sized faces.

The ResNet (Residual Network) network can effectively cope with the problem of gradient vanishing in neural networks [15]. During network training, data are processed through a convolutional layer to produce a new feature map. This relationship between input and output can be viewed as a mapping. One of the roles of the weight parameters in the network is to fit this mapping relationship. In traditional neural networks, it is a difficult task to learn the underlying mapping relation H(x) = x, which represents the input x forming the output after passing through N stacked convolutional layers. However, if the network is designed as H(x) = F(x) + x, i.e., the constant mapping is directly included as part of the network, the problem can be transformed into learning a residual function F(x) = H(x) − x. As long as the residual function F(x) tends to zero, an approximate constant mapping can be composed as H(x) = x.

Figure 3 illustrates the structure of the residual unit, which has an additional path directly connected from the input to the output compared to the normal structure, which is called a jump connection. Through jump connections, the output of the previous layer or layers can be added to the output of this layer and then the result of the summation is passed to the subsequent activation function layer as the final output of this layer, thus improving the training effect of the network. The structure in Figure 3 is defined as in the following equation:(1)$$y=F\left(x,\left\{w_{i}\right\}+x\right)$$
where *x* and *y* are the input and output vectors, respectively, and *w_i_* are the weights. The function *F*(*x*, {*w_i_*}) is the residual mapping to be learned. For the two-layer structure in the figure, the objective function F is shown in the following equation:(2)$$F=W_{2}\sigma \left(W_{1}x\right)$$
where *σ* is denoted as the activation function ReLU (Rectified Linear Unit). In order to ensure that the data can be circulated and processed smoothly, the number of channels here must be consistent, that is, the dimensions of F and *x* must be the same. When their dimensions are not consistent, it is necessary to unify the dimensions by linear projection operation to ensure the accuracy and validity of data processing, as shown in the following equation:(3)$$y=F\left(x,\;\left\{w_{i}\right\}\right)+W_{s}x$$

Unlike conventional target detection, Retinaface’s prediction results are categorized into classification prediction, face frames, and face key point prediction. For any training anchor *i*, the multi-task loss function of the following equation needs to be minimized, as follows:(4)$$L1=L_{cls}\left(p_{i},p_{i}^{*}\right)+\lambda_{1}p_{i}^{*}L_{box}\left(t_{i},t_{i}^{*}\right)+\lambda_{2}p_{i}^{*}L_{pts}\left(l_{i},l_{i}^{*}\right)$$
where *t_i_* and *l_i_* are the predicted bounding box (box), five face key points (landmarks), *t_i_* and *l_i_* are the corresponding truth values, p_i_ is the predicted probability of the face for anchor point *i*, *p_i_^*^* is 1 for positive sample anchors, and 0 is for negative sample anchors. Categorization loss *L_cls_* is the softmax loss function for binary categorization (face/non-face); *L_box_* is the loss function for face box regression; *L_pts_* is the loss function for face key point regression function; *L_box_* is the loss function for face box regression; *L_pts_* is the loss function for face key point regression.

#### 2.1.2. Face Recognition

The algorithm used to extract facial features in InsightFace is ArcFace, which utilizes two types of network backbones: ResNet and MobileFaceNet. The network structure employed in this paper is detailed in Table 1. The input image size was modified from 224 × 224 to 112 × 112, compared to the original ResNet. To accommodate these changes, the convolution kernel sizes and strides of the first and second layers were adjusted. The modified residual units are illustrated in Figure 4. Additionally, the activation function was changed from ReLU (Rectified Linear Unit) to PReLU (Parametric Rectified Linear Unit). The formula for the PReLU function [16] is given in the following equation:(5)$$PReLU\left(x\right)=\left\{\begin{array}{c} x,\;\;\;\;\;x\geq 0\\ a\cdot x,\;\;\;x<0 \end{array}\right.$$
where *x* is the input value and a is a learnable parameter that is usually initialized to a small positive number. The PReLU function introduces a small slope for negative inputs, thus solving the dead neuron problem of the ReLU function.

The main contribution of ArcFace is the proposed ArcFace loss, whose formula is given in the following equation:(6)$$L2=-log\frac{e^{s\left(cos\left(\theta_{y_{i}}+m\right)\right)}}{e^{s\left(cos\left(\theta_{y_{i}}+m\right)\right)+\sum_{j=1,j\neq y_{i}}^{N}e^{scos\theta_{j}}}}$$

Equation (6) is an improved softmax loss function that significantly improves the classification performance by introducing L2 regularization, a scale parameter, and an angular interval m in order to explicitly take into account both the similarity of samples within classes and the variability of samples between classes. The traditional softmax loss function does not explicitly take into account the similarity of intra-class samples and the dissimilarity of inter-class samples when performing the classification task. In contrast, ArcFace normalizes the length of the feature vectors by using L2 regularization, which makes the feature vectors more comparable on the Euclidean space. Meanwhile, by introducing the scale parameter, the scale of the feature vectors can be adjusted, making the similarity of intra-class samples more prominent. In addition, by the setting of the angular interval m, the dissimilarity of the inter-class samples is increased, making the feature vectors between different classes more clearly separated. With these improvements, ArcFace is able to better distinguish between samples from different classes, produce more discriminative feature vector representations, and have higher similarity for within-class samples and greater dissimilarity for between-class samples. This has enabled ArcFace to achieve significant performance gains in face recognition, target recognition, and other classification tasks, making the model more accurate and reliable.

The feature comparison section evaluates the extracted 512-dimensional face features using cosine distance. Cosine similarity is a metric used to quantify the directional similarity between two vectors, which plays an important role in many machine learning and data mining scenarios; the formula is shown in the following equation:(7)$$\cos\theta =\frac{\sum_{k=1}^{n}x_{1k}x_{2k}}{\sqrt{\sum_{k=1}^{n}x_{1k}^{2}}\sqrt{\sum_{k=1}^{n}x_{2k}^{2}}}$$
where *x*_1*k*_ and *x*_2*k*_ are the values of the elements in the two n-dimensional sample points and the range of the angle cosine value is [−1, 1], where 1 means completely similar, −1 means completely opposite, and 0 means irrelevant. This interpretable specific can intuitively understand and explain the degree of similarity between the face feature vectors, so that the recognition results can be better understood and analyzed.

### 2.2. Objection Detection

YOLOv5 is a convolutional neural network-based object detection model and an improved version of YOLOv4 [17], with the model size being approximately one-tenth that of YOLOv4. The model primarily comprises three components: Backbone, Neck, and Prediction. After the image is input, the Backbone network aggregates to form image features at different granularities. Figure 5 illustrates the network structure of YOLOv5s, which includes the Conv module, the C3 module, and SPPF (Spatial Pyramid Pooling Fusion). The Conv module consists of a two-dimensional convolutional layer, a batch normalization layer, and an activation layer concatenated in tandem. The C3 module comprises three convolutional layers and a feature fusion layer, connected by cascades and residuals to enhance feature representation capability. SPPF processes the feature map through multi-scale pooling operations, fuses features of different scales, and generates fixed-length output features, thus improving the model’s feature extraction capability and its ability to handle input images of arbitrary sizes. The Neck network splices and transmits the image features to the Prediction layer, which then generates bounding boxes and prediction categories.

To address the challenges posed by small targets such as hands and mouths, a loss function based on Normalized Wasserstein Distance (NWD), which is a modification of the complete-IoU (CIoU) project [18], is introduced. Given the predicted bounding box and the true bounding box, the loss function is defined as follows:

First, the squared distance between the centers of the predicted bounding box and the true bounding box is defined as
(8)$$d_{center}^{2}={\left(x_{pred}-x_{gt}\right)}^{2}+{\left(y_{pred}-y_{gt}\right)}^{2}+\epsilon$$
where *ϵ* is a small constant to avoid zero values, taken as 1 × 10^−7^.

Next, the distance between the width and height of the predicted bounding box and the real bounding box is defined as
(9)$$d_{wh}=\frac{{\left(w_{pred}-w_{gt}\right)}^{2}+{\left(h_{pred}-h_{gt}\right)}^{2}}{4}$$

Combining the centroid distance and the width–height distance, the squared Wasserstein distance is
(10)$$W_{2}^{2}=d_{center}^{2}+d_{wh}$$

The final loss function is obtained as
(11)$$L_{NWD}=exp\left(-\frac{\sqrt{W_{2}^{2}}}{C}\right)$$
where *C* is a constant, taken as 12.8. The combination of centroid distances and width-height distances makes the NWD-based loss very sensitive to small changes in position and scale, which is crucial for the accurate detection of small targets.

### 2.3. Face Recognition-Based Student Authentication Algorithm

#### 2.3.1. Constraints on the Head–Body Relationship in Frontal-Facing Person Recognition

Let B = [*x_B_*, *y_B_*, *w_B_*, *h_B_*] denote the rectangular box of the body, where *x_B_*, *y_B_* are the coordinates of the upper left corner of the rectangular box, *w_B_* is the width, and *h_B_* is the height. Let H = [*x_H_*, *y_H_*, *w_H_*, *h_H_*] denote the rectangular frame of the human face, where *x_H_*, *y_H_* are the coordinates of the upper-left corner of the rectangular frame, *w_H_* is the width, and *h_H_* is the height.

When the person faces the camera head-on, the face frame H body frame B satisfies the following constraints:

Horizontal Position Alignment: The center of the face should be located roughly at the center level of the body, as shown in the following equation:(12)$$\left|\left(x_{H}+\frac{w_{H}}{2}\right)-\left(x_{B}+\frac{w_{B}}{2}\right)\right|\leq \theta_{x}\cdot w_{B}$$
where *θ_x_* is a scaling factor to adjust the allowable horizontal offset.

Vertical position constraint: The upper boundary of the face should be located in the upper half of the body, i.e.,
(13)$$\left\{\begin{array}{c} y_{H}+h_{H}\leq y_{B}+\frac{h_{B}}{2}\\ y_{H}\geq y_{B} \end{array}\right.$$

Size Proportion: The width of the face should not exceed the width of the body and be of appropriate height, i.e., the constraints shown in the next equation:(14)$$\left\{\begin{array}{c} \frac{h_{H}}{h_{B}}\approx \eta \\ w_{H}\leq w_{B} \end{array}\right.$$
where *η* is the scale factor of the height of the head to the body and, since the scene is a classroom, the scale factor is set to 0.5 here.

#### 2.3.2. Classroom Student Identification Algorithm

The results of identity recognition can be confusing due to angle and overlap; the problem is demonstrated specifically in Figure 6, where a student was double-framed three times. The problem is analyzed for students in the same row with different face sizes depending on the distance. If more than one face appears horizontally in a box (e.g., the green and yellow boxes in Figure 6, the real face should belong to the red box), the identity of the face with the larger size is selected for identification. If more than one face appears vertically in a box, the face information with smaller size and upward position is selected for identification.

Assuming the presence of two rectangular frames, A and B, located horizontally in approximate alignment, the overlapping case occurs when half of the frames overlap. The calculation for the horizontal position of the overlapping case is described by the following equation:(15)$$\left(x_{A}+\frac{w_{A}}{2}\right)-\left(x_{B}+\frac{w_{B}}{2}\right)\leq 0.5w_{A}$$
where *x* and *w* are the coordinates of the x-axis of the upper left corner of the rectangular box and the width of the rectangular box, respectively.

For the vertical case, it is the same as the above equation, as shown in the following equation:(16)$$\left(y_{A}+\frac{h_{A}}{2}\right)-\left(y_{B}+\frac{h_{B}}{2}\right)\leq 0.5h_{A}$$
where y and h are the coordinates of the y-axis of the upper left corner of the rectangular box and the height of the rectangular box, respectively.

In the specific case described in Equation (15), when multiple faces are present in the horizontal position, they are recognized by comparing the sizes of these facial regions. Let the two face rectangular frames be *H*_1_ and *H*_2_; the specific calculation method is shown in the following equation:(17)$$H_{True}=\left\{\begin{array}{c} H_{1},{\;w}_{H_{1}}\cdot h_{H_{1}}\geq w_{H_{2}}\cdot h_{H_{2}}\\ H_{2},{\;w}_{H_{1}}\cdot h_{H_{1}}<w_{H_{2}}\cdot h_{H_{2}} \end{array}\right.$$

The specific process is shown in Algorithm 1, where the input is a list containing information about multiple faces, such as [{‘box’: [448, 332, 511, 434], ‘name’: ‘student1’, ‘score’: 0.785, id: 4031}, {‘box’: [0, 67, 69, 562], ‘name’: ‘student2’, ‘score’: 0.683, id: 4006}, {‘box’: [0, 0, 1280, 875], ‘name’: ‘_IM_INFO’, and ‘score’: 100, id: 99999}], where box is the face coordinates, name is the student’s name, score is the similarity, and anno is the student number. It is important to note that the last element in the list is the image information of the entire human body detection box. The output of the algorithm is a list containing information about a single face. The details of the algorithm are shown in Algorithm 1.
**Algorithm 1:** Student identity verification algorithm based on face recognition**Input**: List of face information containing multiple student identities Lstface**Output**: List of Face Information Containing Individual Student Identities Lstface**1. FUNCTION** Student_Identity_Verify(Lstface)**2.**  filteredFaces <- []**3.**  **FOR** face **IN** Lstface **DO****4.**    x1, y1, x2, y2 <- face[‘box’]**5.**    x1_other, y1_other, x2_other, y2_other <- Lstface[−1][‘box’]**6.**    // Preservation of faces located on the upper part of the body**7.**    **IF** x1 ≥ x1_other **AND** y1 ≥ y1_other **AND** x2 ≤ x2_other **AND** y2 ≤ (0.5 × y2_other) **THEN****8.**      filteredFaces <- filteredFaces + [face]**9.**     **END IF****10.**  **END FOR****11.**  Lstface <- filteredFaces**12.**  **IF LENGTH**(Lstface) > 1 **THEN****13.**    maxArea <- 0**14.**    maxAreaIndex <- 0**15.**    // Preserve the face with the largest area in close proximity**16.**    **FOR** i, face **IN INDEXED** Lstface **DO****17.**     area <- (face[‘box’][2] − face[‘box’][0]) × (face[‘box’][3] − face[‘box’][1])**18.**     **IF** area > maxArea **THEN****19.**      maxArea <- area**20.**      maxAreaIndex <- i**21.**     **END IF****22.**    **END FOR****23.**    Lstface <- [Lstface[maxAreaIndex]]**24.**  **END IF****25.**  **RETURN** Lstface**26.**
**END FUNCTION**

## 3. Case Study

### 3.1. Experimental Platforms

Face recognition experimental environment: OS: Ubuntu22.04.2 LTS installed in WSL2 (Windows Subsystem for Linux 2); GPU: NVIDIA GeForce RTX 2080 Ti; Language: Python3.11.4, torch2.1.0; Acceleration environment: cuda12.2.

Image classification and object detection experimental environment: OS: Windows 10 Professional Edition 22H2; GPU: NVIDIA GeForce RTX 2080 Ti; Language: Python3.8.0, torch1.9.1; Acceleration environment: cuda12.2.

### 3.2. Dataset

#### 3.2.1. Classroom Face Dataset

The dataset used for the experiments consists of the publicly available dataset CASIA-Webface [19] and self-collected face images from the classroom site. The dataset was collected from classroom scenes as shown in Figure 7. The CASIA-WebFace dataset contains a total of 494,414 face images from 10,575 different identities. The dataset contains face images of different ages, genders, ethnicities, and expressions. For the self-collected images, they were first cropped to the size of 112 × 112 pixels required for training and then the cropped images were data-enhanced by conventional methods such as flipping them left and right, adding noise, and adjusting brightness. In order to meet the needs of classroom face recognition, it is not enough to use only frontal photos for conventional data enhancement. Each photo also needs to contain separate faces with different angles in order to learn enough information to be able to recognize faces with different poses in subsequent training. It is also necessary to ensure that five key parts of the image, such as the left and right eyes, the tip of the nose and the left and right corners of the mouth, are clearly visible. To achieve this, face pose enhancement methods [20,21] are used to simulate poses such as head up, head down, and head turn. Examples of images after data enhancement are shown in Figure 8, where the image above shows, from left to right, the cases of raising the head 45°, 30°, and 15° and lowering the head 15°, 30°, and 45°, respectively, and the following image shows, from left to right, the cases of turning the head 45°, 30°, and 15° to the left and turning the head 15°, 30°, and 45° to the right. By applying the data enhancement method in Figure 8 to the captured real classroom images, a total of 31458 face images for 99 categories were finally obtained and an example of the images in the captured classroom face dataset is shown in Figure 9.

#### 3.2.2. Classroom Behavior Dataset

The classroom behavior dataset is divided into two types: image classification and object detection.

Image Classification Dataset

The image data for classroom behaviors were obtained from the web and the classroom site and images of individual students were cropped out for collection to allow for better focus on student behaviors during training. In total, the dataset contains the following behaviors along with their respective labels and number of samples: looking up (lookup, 1760), looking down (lookdown, 1228), using a cell phone (usephone, 392), reading and writing (write, 577), sleeping (sleep, 232), and yawning (yawn, 346). The state explanation for each behavior is shown in Table 2 and the sample example is Figure 10.

2.Object Detection Dataset

In the object detection dataset based on head-up and head-down movements, hand and mouth positions are explicitly labeled as auxiliary labels. Specifically, these annotations cover three contexts: an open mouth when yawning, hand-pen contact when writing, and hand-phone contact when using a cell phone. Table 3 shows the specifics of the behavioral definitions after the use of auxiliary labels. Figure 11 provides annotation examples for object detection.

Based on the aforementioned labeling methods, a total of 1609 images were labeled, forming the base dataset for the experiment. To enhance the dataset’s diversity and robustness, various image enhancement techniques were applied, including the addition of Gaussian noise, pretzel noise, and random brightness and contrast transformations. After completing data preprocessing, the dataset was randomly divided into a training set and a validation set using an 8:2 ratio, ensuring the independence and validity of the model training and validation processes.

Next, five mainstream YOLO object detection models—YOLOv5s, YOLOv7-tiny, YOLOv8s, YOLOv9s, and YOLOv10s—were selected for the training task. To ensure consistency in experimental conditions, the training period for all models was uniformly set to 120 epochs to facilitate performance comparison. Since the models are intended for deployment on embedded devices, considerations also included the number of model parameters, computation requirements, model weight file sizes, and compatibility.

### 3.3. Model Training

#### 3.3.1. Face Recognition Model Training

The network structure of R18, R34, R50, R100, and Mobileface were used for model training, respectively. The network training uses asynchronous stochastic gradient descent with a momentum term of 0.9; the initial learning efficiency of the weights is 0.02, the decay coefficient is set to 0.0005, the number of samples in each training batch is 128, the total number of training rounds is 20, and the dimension of the face feature embedding vectors is 512.

#### 3.3.2. Training of the Classroom Behavior Classification Model

The dataset was divided into training and validation sets according to the ratio of 8:2 and the training set was augmented with data using random rotation, random luminance-contrast transformation, and random noise addition before training, resulting in 6585 samples. A classification model with a similar number of scale parameters as yolov5s-cls was used for training.

### 3.4. Embedded Device

The embedded device is chosen to be the Atlas 200 DK, which is an edge AI development kit from Huawei. It is equipped with 8 GB of RAM and powered by Huawei’s in-house developed AI processor, the Ascend 310, which has powerful neural network computation and highly parallel processing capabilities. The Atlas 200 DK also supports a variety of commonly used programming languages and development frameworks, such as C++, Python, Pytorch, and TensorFlow, making it easy for developers to develop on-demand. The Atlas 200 DK and its interfaces and hardware are shown in Figure 12.

### 3.5. Program Design

Figure 13 specifically describes the logic for determining student behavior based on the results of target detection. The process has two parts: first, to determine whether the labels “usephone”, “write”, and “yawn” for hand or mouth movements are detected, and then to determine student behaviors such as cell phone use or yawning based on their containment relationship with head-up and head-down, as shown in Figure 11. The detected head-up and head-down boxes will contain the boxes corresponding to cell phone use and yawning. In order to measure the degree of overlap between two rectangular boxes, a new metric called “Intersection over Smallest Bounding Area (ISBA)” was introduced. This metric calculates the area of the intersection of two rectangles and divides it by the area of the smaller of the two rectangles. This method gives a fraction of the coverage of the small rectangle relative to the larger rectangle. ISBA is calculated as follows:(18)$$ISBA=\frac{BodyResut\cap OtherResult}{OtherResult}$$
where BodyResult represents the large rectangular box for both head-up and head-down cases and OtherResult represents the small rectangular box for hand or mouth.

### 3.6. Application Development

The operating system burned in Atlas 200 DK is Ubuntu 18.04.6 Servers. The application development is based on Python 3.8.5 using the CANN (Compute Architecture for Neural Networks) development tool with version Ascend-cann-toolkit_6.0.0.alpha001_linux-aarch64 to extract the facial features of 79 students’ photo IDs with bare-chested solid color background and store them in the sqlite database.

The developed application mainly involves image/video data processing, model inference, and data transmission and requires numpy, opencv, and sqlite libraries in addition to the Python language API library provided by Huawei’s pyACL (Python Ascend Computing Language). The developed code directory includes python script files and a data directory, scripts directory and model directory, where the data directory stores the database files of student information and the tag files of YOLOv5, the model directory stores the model files, and the scripts store the bash scripts of the camera calls, network communication, and so on.

The development application has to perform pyACL initialization, apply for operation management resources, transfer data, and perform model inference. After all the data processing is finished, the operation management resources are to be released and finally, the execution of pyACL de-initialization follows, as shown in Figure 14.

#### 3.6.1. Model Optimization

In order to improve the inference efficiency of the Rise AI processor, the network models built using the open-source framework need to be converted by the ATC (Ascend Tensor Compiler) tool to generate offline models (*.om files) adapted to the Rise AI processor.

The input sizes and input names of the models were observed using the Netron model visualization tool and the input sizes of the two onnx files for face detection and face recognition were 1 × 3 × ? × ? and None × 3 × 112 × 112. Based on debugging, it was determined that the input dimensions for the two models are 1 × 3 × 640 × 640 and 1 × 3 × 112 × 112. These dimensions represent 1 image, 3 channels, and image sizes of 640 × 640 and 112 × 112, respectively. onnx model files are converted to om model files with the following commands:

atc --model=det_10g.onnx --framework=5 --input_shape=“input.1:1,3,640,640” --input_format=ND --output= det_10g --soc_version=Ascend310

atc --model=R100_WEBF.onnx --framework=5 --input_shape=“data:1,3,112,112” --output= R100_WEBF --soc_version=Ascend310

For the model format provided by YOLOv5 is pt, it must be converted to an onnx file first and then to an om file; the conversion commands are as follows:

python export.py --weights yolov5s.pt --opset 12 --simplify --include onnx

atc --input_shape=“images:1,3,640,640” --input_format=NCHW --output=“yolov5s” --soc_version=Ascend310 --framework=5 --model=“yolov5s.onnx” --output_type=FP32 

atc --model=yolov5s-cls.onnx --framework=5 --input_format=NCHW --input_shape=“images:1,3,224,224” --output=“yolov5s-cls” --output_type=FP32 --soc_version=Ascend310

#### 3.6.2. Debugging Process

Adjusting the cosine distance threshold has importance in face recognition. Figure 15 shows a screenshot of the runtime logs with an initial threshold value of 0.28 and a serious misdetection problem when performing face recognition. By analyzing the logs of the debugging process, it was found that the similarity (sim) values at the time of misdetection were distributed between 0.28 and 0.36. The selected green portion of Figure 15 represents the misidentified identity, indicating that lower thresholds resulted in mistaking faces with lower similarity as matches. In order to solve the misdetection problem, it was decided to raise the threshold value to 0.38 to improve the accuracy and reliability of the recognition.

### 3.7. On-Site Testing of the Embedded Device

To enhance the evaluation of the embedded device’s application in a genuine classroom setting, a test was conducted in a different class that was not included in the training set. The specific testing conditions are outlined in Table 4. Throughout the test, the Atlas 200 DK was positioned on a stable support system located at the front of the classroom. The log file will record additional test results apart from the images, including the detection time, student ID, student’s location coordinates, and confidence level for each current image.

## 4. Results Analysis

### 4.1. Model Training Result

#### 4.1.1. Face Recognition Results

Using LFW [22], CFP [23], and AgeDB-30 [24], three face recognition commonly used test sets, the training results are shown in Table 5.

Specifically, it can be seen that on the LFW dataset, the R100 network structure achieves the highest accuracy rate of 99.31%. On the CFP and AgeDB datasets, R100 also achieved the highest accuracy rate, of 94.28% and 93.31%, respectively. With more layers and stronger expressive ability than other structures, ResNet-100 is able to better capture the features of face images, thus achieving more accurate face recognition results. In order to ensure accurate face recognition in complex classroom scenarios and to fully utilize the powerful computational capabilities of the Atlas 200 DK, ResNet-100 was chosen to be deployed as the training model.

#### 4.1.2. Behavioral Recognition Results

The training results for image classification and object detection are, respectively, shown in Table 6 and Table 7. From the training results, both methods achieved better results but the actual results need to be further analyzed. Looking at the training results as a whole, the model performs well on most of the categories, with a precision between 0.924 and 0.964 and a mAP50 between 0.904 and 0.97, the model’s prediction for these categories is more accurate and stable. However, the recall of the categories “sleep” and “write” is low, at 0.867 and 0.831, respectively, so the model may miss some samples of sleeping and writing.

### 4.2. Comparison of Different Labeling Scenarios

Figure 16a shows the confusion matrix for image classification, where it can be seen that the diagonal portion has been rendered in the darkest color, but there is still some confusion. The model has confusion between the three categories of “uesphone”, “lookdown”, and “write”, due to the neglect of hand features. Secondly, there was a significant confusion observed between the “yawn” behavior and the actions of “lookup”, “lookdown”, and “read” and “write”. This confusion can be attributed to the model’s inadequate attention to the student’s oral features.

According to Figure 16b, it can be seen that, due to the complexity of the scene, it appears that more backgrounds are predicted as “lookup” and “lookdown”. The confusion between “write”, “yawn”, “usephone”, “lookup”, and “lookdown” observed in Figure 16a is mitigated due to the annotation method. However, a certain level of confusion arises between the labels “sleep” and “lookdown”. Figure 17 shows the behavior recognition effect of yolov5, which is better for head up and head down but better for the recognition of cell phone playing and writing behaviors in the distance is still missing. According to the recognition results, the confusion generated by the background is shown by the students who only show part of their limbs but not their faces.

Table 8 details the performance of the evaluated object detection models on various key metrics. Regarding the number of model parameters, computational complexity (measured in GFLOPs), and model size, YOLOv5s demonstrates a significant efficiency advantage. It has the second-lowest number of parameters, only surpassed by YOLOv7-tiny, and is much lower than YOLOv8s and YOLOv9s. This characteristic suggests that YOLOv5s is highly applicable in resource-constrained environments. Further analyzing the performance, YOLOv5s excels in precision and mean average precision (mAP50), with its performance close to the optimal YOLOv8s and YOLOv10s models, indicating strong detection capability. Additionally, the recall (Recall) of YOLOv5s is maintained at a high level (0.914), ensuring the integrity of detection results. In summary, YOLOv5s effectively reduces computing resource consumption and storage requirements while maintaining high precision and high recall, making it an ideal choice for balancing performance and efficiency. It is also worth noting that while newer models may be technologically advanced, they face deployment challenges on the Atlas 200 DK, further emphasizing the value of YOLOv5s as a mature solution.

### 4.3. Embedded Device Test Results

The green boxed portion in Figure 18 shows the recognition results using Algorithm 1, which has been able to identify the student correctly.

Figure 19 shows the results generated by testing the Atlas 200 DK in a classroom, showing the overall status of student behavior in the classroom, with red rectangles framing selected student body parts. When negative behaviors are detected, the last four digits of the student’s school number, the student’s behavior, and the confidence level are indicated in the upper right corner of the rectangle and unidentified students and positively behaving students are marked with “no info!”. In addition, information about overall student behavior in the classroom, individual students with negative behaviors, and current timestamps will be stored in the database, providing a basis for further analysis, evaluation, and improvement in classroom teaching strategies and individualized instruction for students.

The process of identifying the student’s identity and location using face recognition and target detection algorithms and uploading the body region images to the server is as follows:

First, face detection and target detection are performed in parallel. The face recognition algorithm is used to determine the identity and location frame of the face while the target detection algorithm is used to detect the location frame of the body. Next, the face position frame obtained by the face recognition algorithm and the human body position frame obtained by the target detection algorithm are matched to determine whether they belong to the same person. When the match fails, the person is ignored and the process continues to the next person. When the matching is successful, it means that the person exists in the database and the image region corresponding to the human body position axis can be extracted. Finally, the extracted axis in the human body box is uploaded to the server.

Table 9 shows the test results of object detection in real classroom scenarios. The precision for the behavior categories of “lookdown”, “lookup”, and “sleep” reached over 0.78, while the precision for smaller target behaviors such as “usephone”, “write”, and “yawn” was relatively low. The recall shows that small targets have a higher miss-detection rate compared to large targets. Using the same image on a computer and Atlas 200 DK, the poorer detection results of Atlas 200 DK are due to the two conversions of the model file from pt format to om format.

The average false detection rate of face recognition is 41.5%, the minimum missed detection rate is 10%, and the maximum false detection rate is 93% during the whole test; the highest false detection rate is when the number of students with their heads down is the highest. In addition to students looking down, touching and covering their faces with their hands is also a cause of false detection. When students look down, touch their faces with their hands, or cover their faces, their facial features are distorted or partially or fully occluded, making it difficult for the algorithm to recognize them accurately, which increases the likelihood of a false detection. Although the head pose simulation enhancement method was used in creating the dataset, the data of different angles and occluded faces in the dataset are still insufficient. However, in the classroom, the position of the students does not change much, so the recognition results in the same area can be continuously recorded for identification purposes.

Based on the results log, Figure 20 shows the distribution of overall student behavior in the class generated from the results of the 10-min classroom test. The figure indicates that the two primary categories, “lookdown” and “lookup”, had a relatively high distribution, while the categories of “using phone”, “write”, and “yawn” were less frequent, which also reflects the poorer detection results for smaller target behaviors.

Figure 21 shows the time (in 1-min intervals) on the x-axis and the student action categories on the y-axis. For this experiment, three students with different seating positions in a practice session were selected as the subjects. The accuracy of the model was validated by observing the fit between the predicted student action curves and the actual student action curves. The results show that the predictions were most accurate for the student sitting in the front row (image size: 271 × 175), with an accuracy of 0.911, followed by the student in the middle (image size: 200 × 110), with an accuracy of 0.778, and the student in the last row (image size: 113 × 78) had the poorest prediction accuracy of 0.577.

The Atlas 200 DK will temporarily record behavioral data in the classroom and transmit the data to the cloud at the end of the class. As shown in Figure 22, a management system developed in Django records students who exhibit negative behaviors. In the exercise class, looking up is considered a negative behavior. When the duration and frequency of looking up exceed a certain threshold, it is recorded in the database. For example, the student shown in the figure has the last four digits of their student ID as 4028.

## 5. Conclusions

This research aims to enhance the accuracy of individual student analysis in a classroom setting and lay the foundation for personalized instruction. Utilizing face recognition technology and target detection algorithms, we successfully implemented student identification, behavior detection, and embedded application development. We expanded the face dataset by collecting classroom face data and applying data enhancement methods, selecting the most accurate face recognition model after training and comparison. We expanded the classroom behavior dataset by collecting classroom images and applying data enhancement methods. After training and comparing image classification and object detection methods, we selected the object detection algorithm with superior accuracy and ease of deployment on embedded devices. After using the YOLOv5 algorithm to locate the body regions of students exhibiting various behaviors, we combined it with the face recognition algorithm to identify students by correlating body and face regions. Finally, we designed and developed an application on the embedded device Atlas 200 DK, successfully intercepting and identifying individual student classroom image sequences and uploading them to a server for management, accessible through a Django-developed interface. 

However, there are some limitations in our approach and the detection performance of the model is not very satisfactory, especially for long-distance and small targets. Future work can focus on expanding the dataset and improving the model. Additionally, engaging with the Huawei Ascend community can help us better adapt to the latest technology. In terms of privacy, the system can be further improved to better personalize instruction by showing individual students’ behavioral profiles only to themselves and by collecting and storing only the necessary log data while reducing image data to minimize privacy risks. Furthermore, one of our ongoing efforts is to achieve more accurate classroom analysis in multimodal directions by incorporating sound sensors on the Atlas 200 DK. Future research could explore more multifaceted applications by processing and analyzing intercepted classroom image sequences and log data. For example, researchers could utilize rich student data to drive further studies in educational psychology and pedagogy and teachers could use this technology to gain detailed insights into individual students’ behavior and learning patterns, leading to more targeted teaching methods.

## Figures and Tables

**Figure 1 sensors-24-04582-f001:**
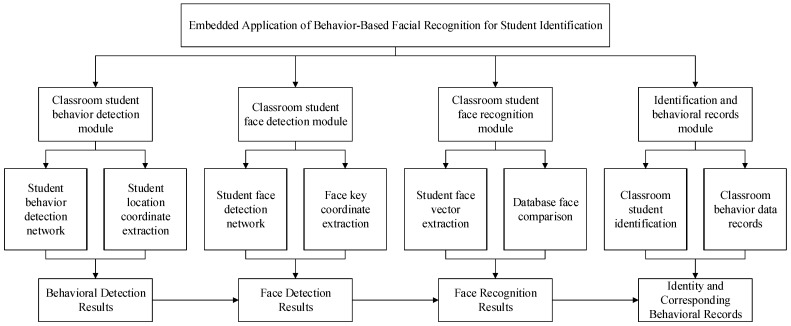
System architecture.

**Figure 2 sensors-24-04582-f002:**
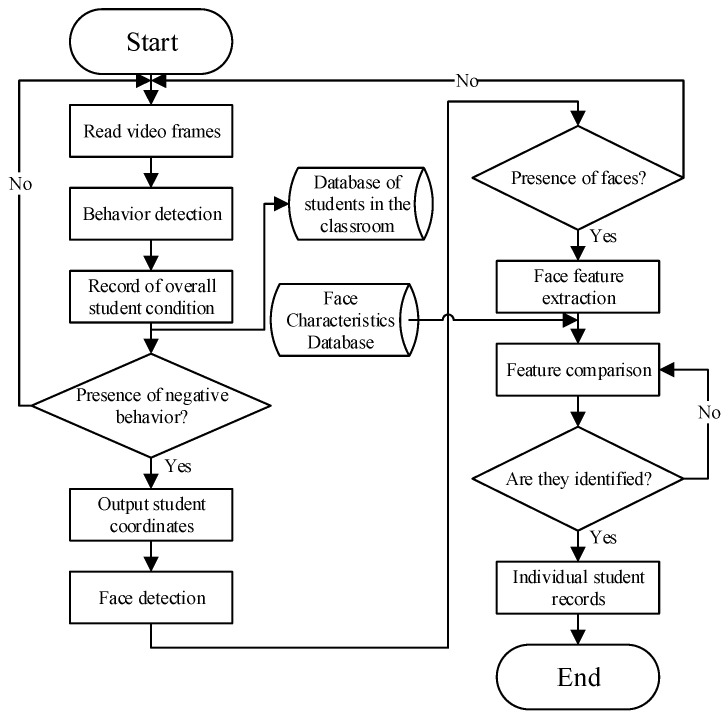
System algorithm flow.

**Figure 3 sensors-24-04582-f003:**
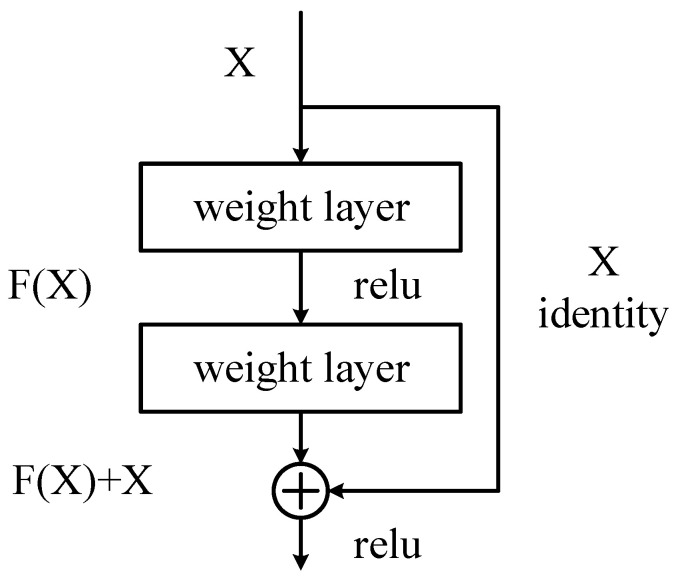
Residual unit structure.

**Figure 4 sensors-24-04582-f004:**
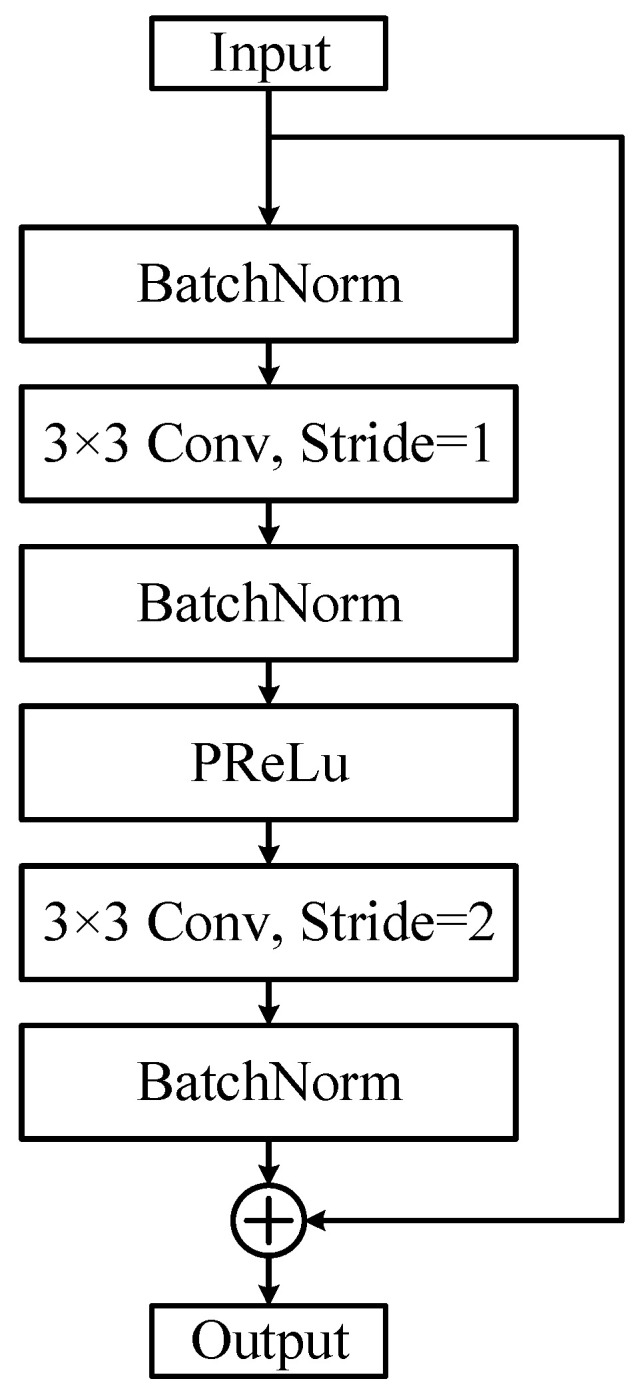
The improved residual unit.

**Figure 5 sensors-24-04582-f005:**
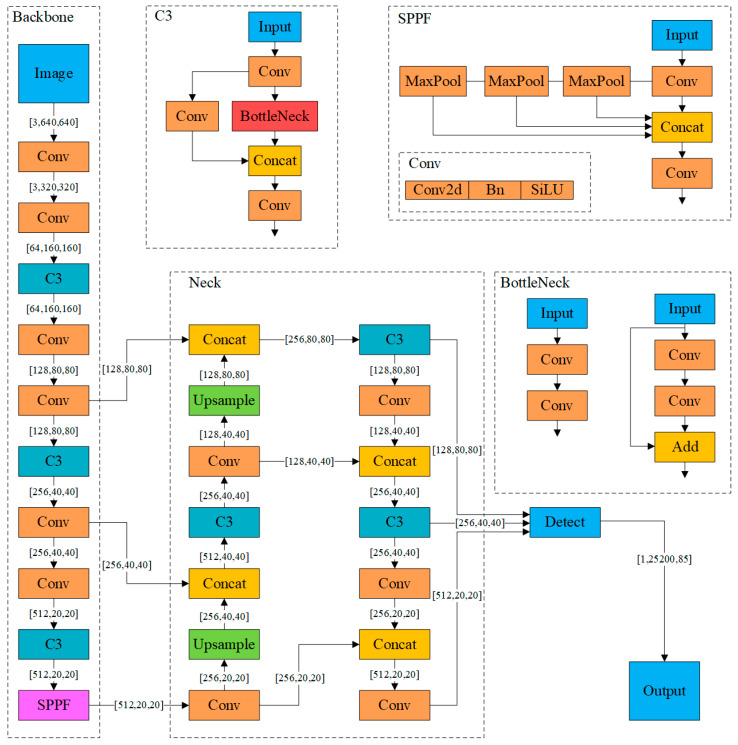
Network structure of YOLOv5s.

**Figure 6 sensors-24-04582-f006:**
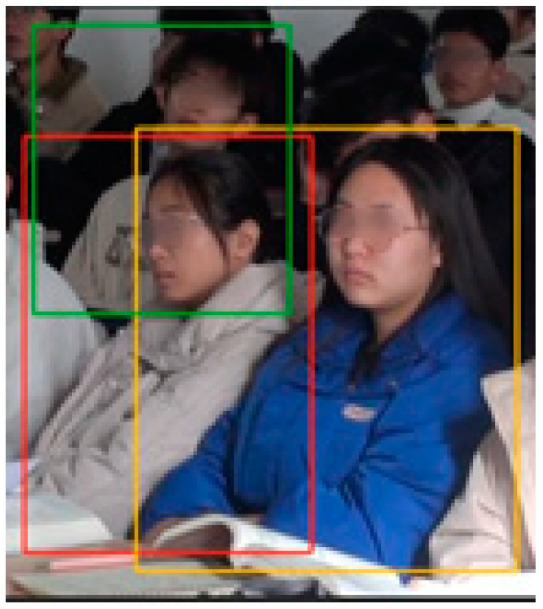
A student was boxed multiple times.

**Figure 7 sensors-24-04582-f007:**
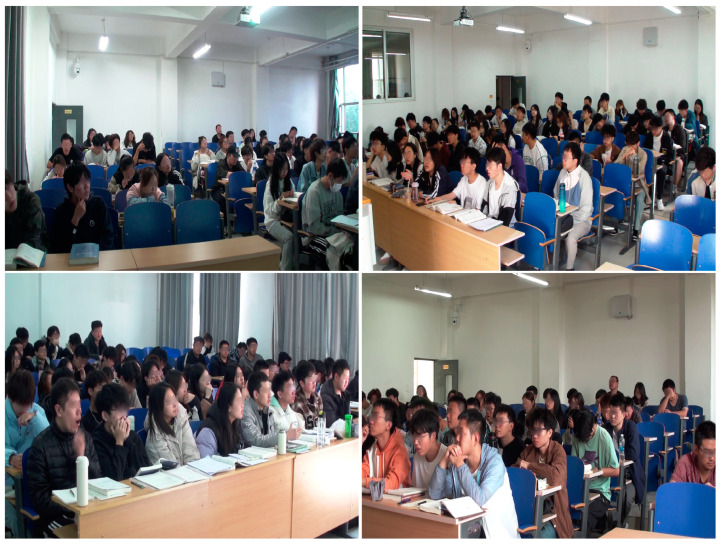
Classroom scenario.

**Figure 8 sensors-24-04582-f008:**
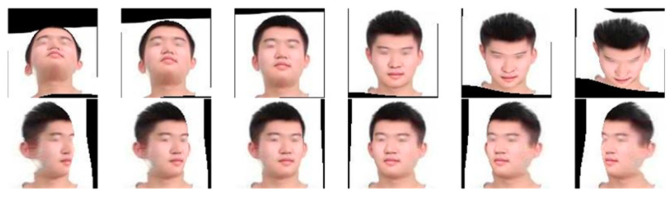
Example of image after data augmentation.

**Figure 9 sensors-24-04582-f009:**
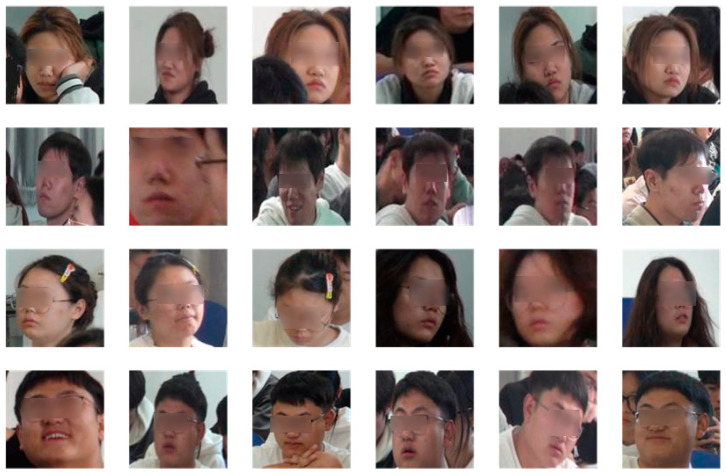
Example of classroom face dataset.

**Figure 10 sensors-24-04582-f010:**
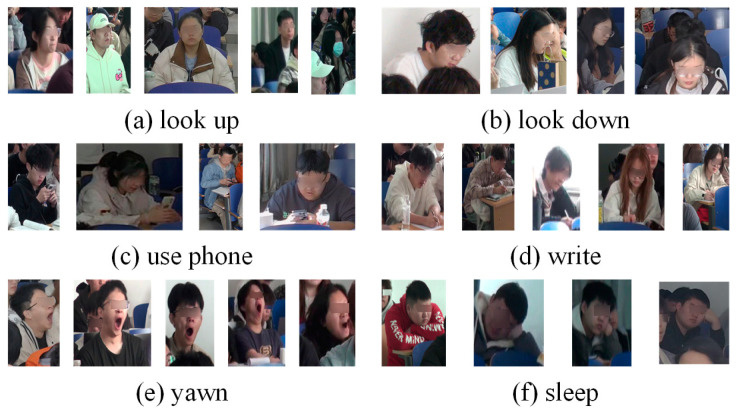
Classroom behavior dataset for image classification.

**Figure 11 sensors-24-04582-f011:**
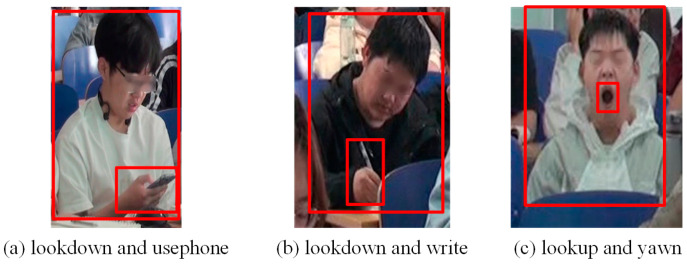
Annotation examples for object detection.

**Figure 12 sensors-24-04582-f012:**
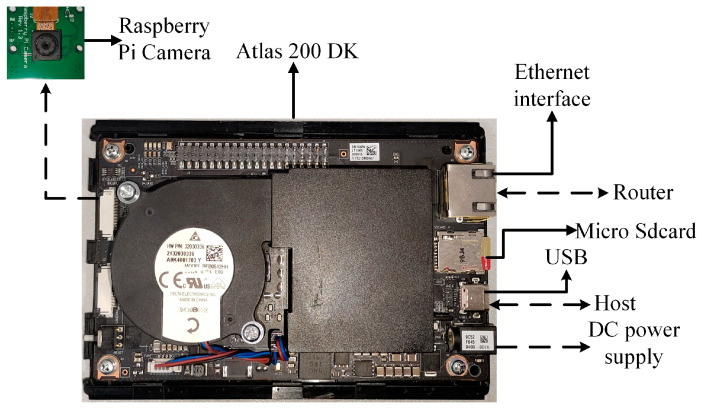
Atlas 200 DK and its interfaces and connections.

**Figure 13 sensors-24-04582-f013:**
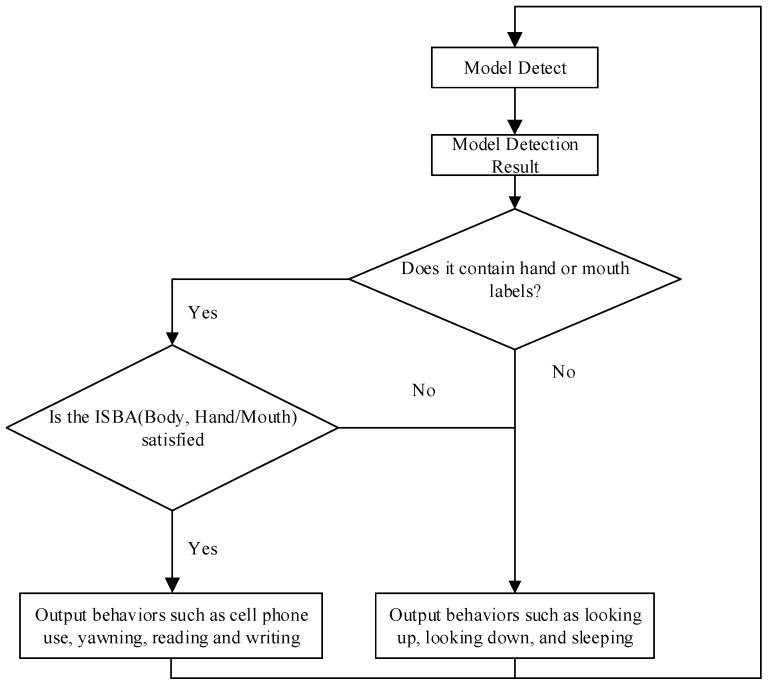
Behavior judgment process.

**Figure 14 sensors-24-04582-f014:**
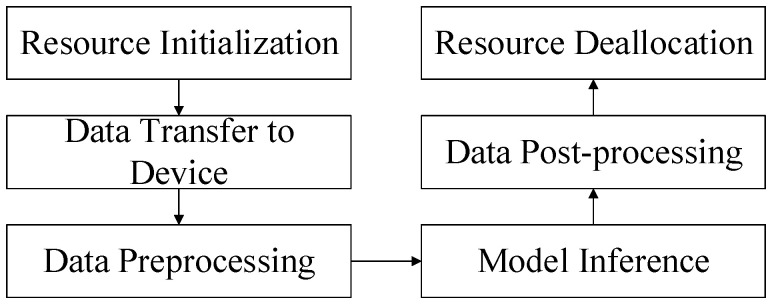
Application implementation process.

**Figure 15 sensors-24-04582-f015:**
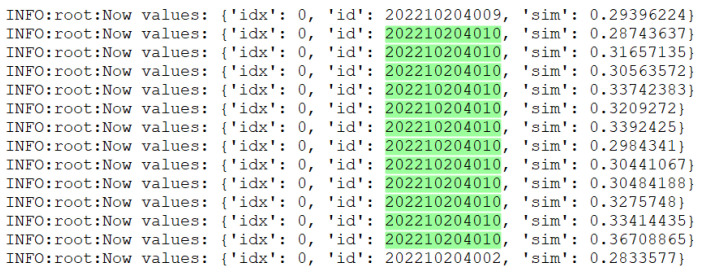
Runtime logs.

**Figure 16 sensors-24-04582-f016:**
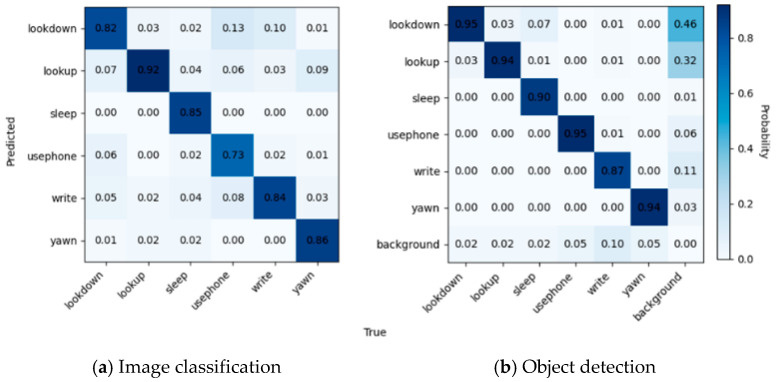
Confusion matrices for image classification and object detection.

**Figure 17 sensors-24-04582-f017:**
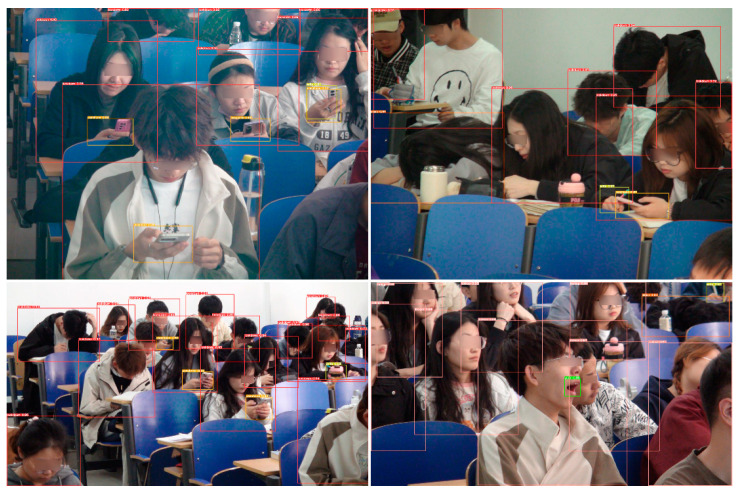
Detection results.

**Figure 18 sensors-24-04582-f018:**
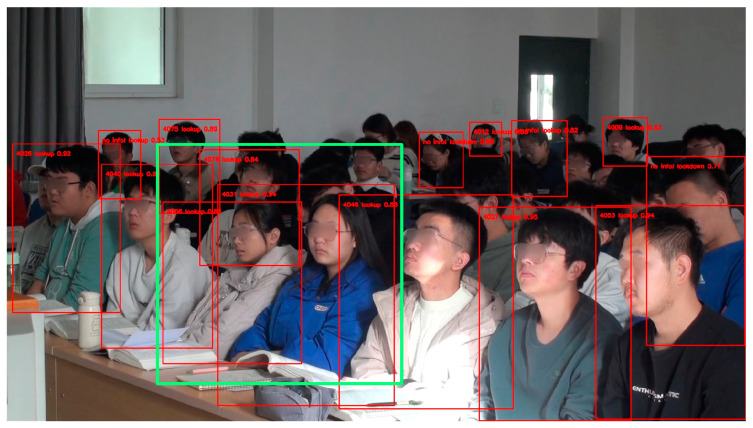
The identity of the students in the green box can be correctly identified.

**Figure 19 sensors-24-04582-f019:**
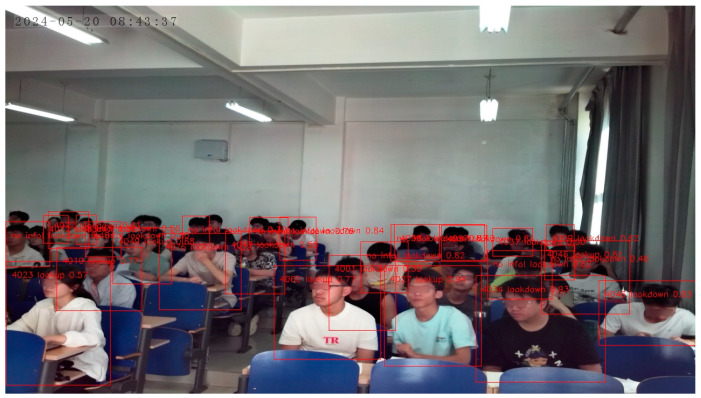
Example of student image results by Atlas 200 DK.

**Figure 20 sensors-24-04582-f020:**
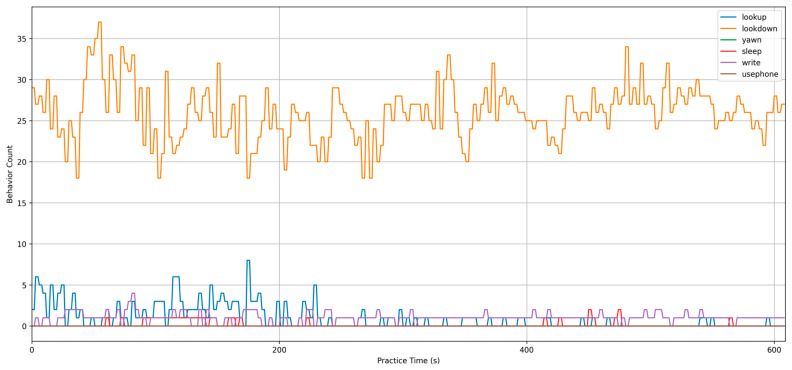
Distribution of overall student behavior in the class.

**Figure 21 sensors-24-04582-f021:**
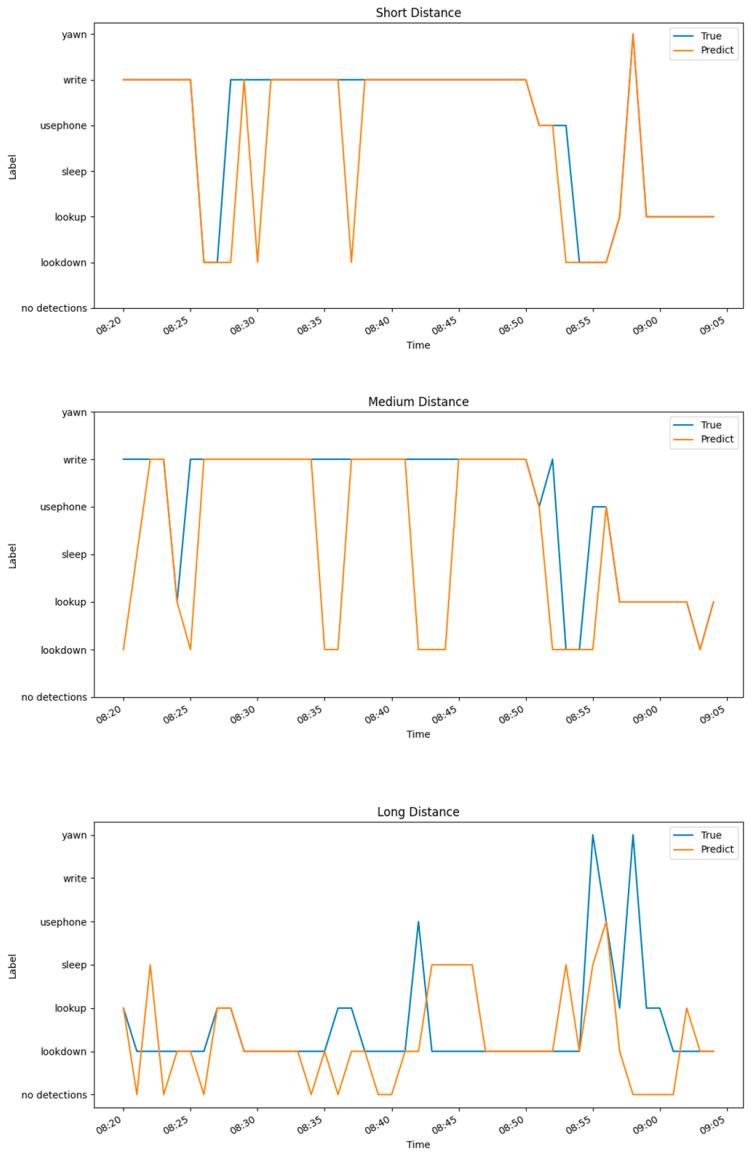
Behavioral timing chart for three students with different distances.

**Figure 22 sensors-24-04582-f022:**
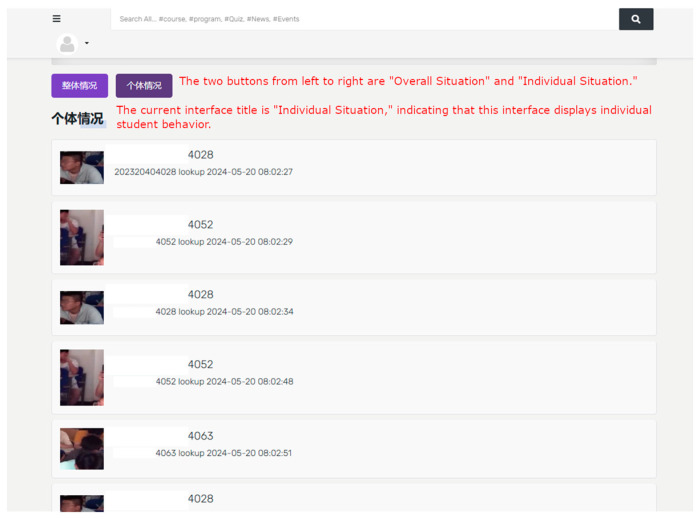
System interface displaying individual student behavior.

**Table 1 sensors-24-04582-t001:** ResNet network parameters used in Insightface.

Layer Name	Output Size	18-Layer	34-Layer	50-Layer	100-Layer
Conv1	112 × 112	3 × 3, 64, stride 1			
Conv2_x	56 × 56	3 × 3 max pool, stride 2			
		$\left[\begin{array}{cc} 3\times 3, & 64\\ 3\times 3, & 64 \end{array}\right]\times 2$	$\left[\begin{array}{cc} 3\times 3, & 64\\ 3\times 3, & 64 \end{array}\right]\times 3$	$\left[\begin{array}{cc} 1\times 1,\; & 64\\ 3\times 3, & 64\\ 1\times 1, & 256 \end{array}\right]\times 3$	$\left[\begin{array}{cc} 1\times 1,\; & 64\\ 3\times 3, & 64\\ 1\times 1, & 256 \end{array}\right]\times 3$
Conv3_x	28 × 28	$\left[\begin{array}{cc} 3\times 3, & 128\\ 3\times 3, & 128 \end{array}\right]\times 2$	$\left[\begin{array}{cc} 3\times 3, & 128\\ 3\times 3, & 128 \end{array}\right]\times 4$	$\left[\begin{array}{cc} 1\times 1,\; & 128\\ 3\times 3, & 128\\ 1\times 1, & 512 \end{array}\right]\times 4$	$\left[\begin{array}{cc} 1\times 1,\; & 128\\ 3\times 3, & 128\\ 1\times 1, & 512 \end{array}\right]\times 4$
Conv4_x	14 × 14	$\left[\begin{array}{cc} 3\times 3, & 256\\ 3\times 3, & 256 \end{array}\right]\times 2$	$\left[\begin{array}{cc} 3\times 3, & 256\\ 3\times 3, & 256 \end{array}\right]\times 6$	$\left[\begin{array}{cc} 1\times 1,\; & 256\\ 3\times 3, & 256\\ 1\times 1, & 1024 \end{array}\right]\times 6$	$\left[\begin{array}{cc} 1\times 1,\; & 256\\ 3\times 3, & 256\\ 1\times 1, & 1024 \end{array}\right]\times 23$
Conv5_x	7 × 7	$\left[\begin{array}{cc} 3\times 3, & 512\\ 3\times 3, & 512 \end{array}\right]\times 2$	$\left[\begin{array}{cc} 3\times 3, & 512\\ 3\times 3, & 512 \end{array}\right]\times 3$	$\left[\begin{array}{cc} 1\times 1,\; & 512\\ 3\times 3, & 512\\ 1\times 1, & 2048 \end{array}\right]\times 3$	$\left[\begin{array}{cc} 1\times 1,\; & 512\\ 3\times 3, & 512\\ 1\times 1, & 2048 \end{array}\right]\times 3$

**Table 2 sensors-24-04582-t002:** Student’s behaviors and their states.

Behavior	State
look up	Head up and in line with body direction
look down	Head tilt down
use phone	Head tilted down and hands in contact with cell phone
write	Sitting upright with hands in contact with books and pens
sleep	Arms support the head and eyes are closed.
yawn	Mouth wide open or covered with hand at the same time

**Table 3 sensors-24-04582-t003:** Improved student’s behaviors and their labels.

Behavior	Labels
look up	lookup
look down	lookdown
use phone	lookdown, usephone
write	lookdown, write
sleep	sleep
yawn	lookdown or lookup, yawn

**Table 4 sensors-24-04582-t004:** Image classification training results.

Condition	Description
Number of Facial Features in the Database	79
Maximum Number of Faces Exposed in Real Scene	39
Duration of the Course	45 min
Number of Course Instances	2
Raspberry Pi Camera Parameters	5 million pixels
Image Resolution	1280 × 720

**Table 5 sensors-24-04582-t005:** Training results.

Network Architecture	Accuracy-Highest (%)		
	LFW	CFP	AgeDB-30
R18	99.16	91.15	92.06
R34	99.15	92.77	92.66
R50	99.21	93.58	92.26
Mobileface	99.20	93.61	92.70
R100	99.31	94.28	93.31

**Table 6 sensors-24-04582-t006:** Image classification training results.

Name	Accuracy
Alexnet [25]	0.649
Yolov5s-cls	0.865
Resnet-34 [15]	0.850
Mobilenet	0.824
efficientNet-B0 [26]	0.814
ConvNeXt-T [27]	0.848

**Table 7 sensors-24-04582-t007:** Object detection training results.

Class	Precision	Recall	mAP50
lookdown	0.932	0.927	0.97
lookup	0.941	0.924	0.968
sleep	0.964	0.867	0.963
usephone	0.924	0.918	0.951
write	0.945	0.831	0.904
yawn	0.952	0.933	0.939

**Table 8 sensors-24-04582-t008:** Object detection training results.

Model	Precision	Recall	mAP50	Parameters	GFLOPs	Size (MB)
YOLOv5s	0.938	0.914	0.961	7,026,307	15.8	14.3
YOLOv7-tiny	0.894	0.849	0.907	6,028,518	13.2	12.3
YOLOv8s	0.971	0.945	0.966	11,127,906	28.4	22.5
YOLOv9s	0.939	0.892	0.944	97,447,236	39.6	20.3
YOLOv10s	0.965	0.913	0.958	8,039,604	24.5	16.5

**Table 9 sensors-24-04582-t009:** Object detect test results of real scenarios.

Class	Precision	Recall
lookdown	0.791	0.785
lookup	0.818	0.803
sleep	0.783	0.628
usephone	0.697	0.428
write	0.674	0.514
yawn	0.678	0.62

## Data Availability

The data presented in this study are available on request from the corresponding author. The data are not publicly available due to privacy restrictions.

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
