# Peer review of "Deep Visual Computing of Behavioral Characteristics in Complex Scenarios and Embedded Object Recognition Applications"

_sensors, 2024, doi:10.3390/s24144582_

Round 1

Reviewer 1 Report

Comments and Suggestions for Authors

Comments and suggestions for authors

This article primarily investigates methods for analyzing student behavior and conducting individual identification in classroom environments using artificial intelligence technology. The paper emphasizes the importance of leveraging artificial intelligence and big data to enhance teaching quality and points out that existing studies tend to focus on assessing student groups, often neglecting personalized teaching support for individual students. The authors propose an integrated system based on deep learning, which includes modules for facial detection, facial recognition, and behavior detection, providing comprehensive technical support for classroom behavior analysis and demonstrating a certain level of innovation and contribution. However, further improvements are needed in aspects such as algorithm details and experimental analysis:

1. It is recommended that the abstract include key statistical data or result metrics to support the research findings.

2. In the introduction section, it is suggested to elaborate on the limitations of existing facial recognition research and how this study addresses those gaps. The authors should also clearly articulate the innovations or contributions of this study to facilitate a clearer understanding for readers.

3. In the section on algorithm principles, a more detailed description of the key technologies and algorithms employed is needed, including the rationale for algorithm selection, details of the model architecture, and settings of hyperparameters.

4.The authors merely mention the algorithms used in the manuscript, but these networks lack detailed descriptions, such as the specific parameters of each layer, the manner in which layers are connected, and the input and output dimensions of the networks. It is also recommended to add an architectural diagram of the model algorithms.

5. In the experimental section, the authors have not compared their work with the latest technologies or conducted performance comparisons under the same conditions. Additionally, the document does not explicitly state the specific advantages and disadvantages of the methods used in comparison to others.

6. In the experimental section, the conclusion at line 529 is drawn that " and the highest false detection rate is when the number of students with their heads down is the highest. In addition to students looking down, touching and covering their faces with their hands is also a cause of false detection.", please provide a corresponding explanation for this conclusion.

7. The article conducted a system evaluation in a real classroom environment, so in the conclusion section, the paper should discuss the potential challenges that may be encountered when deploying the model in an actual classroom setting, such as real-time performance, device compatibility, and user privacy. This will help to enhance the credibility of the research and can provide guidance for future research and development efforts.

Author Response

Comments 1: It is recommended that the abstract include key statistical data or result metrics to support the research findings.

Response 1: Thank you for pointing this out. We agree with this comment. Therefore, we have added key statistical data and result metrics to the abstract to more clearly support the research findings. This change can be found in the revised manuscript on page 1, paragraph 1.

Comments 2: In the introduction section, it is suggested to elaborate on the limitations of existing facial recognition research and how this study addresses those gaps. The authors should also clearly articulate the innovations or contributions of this study to facilitate a clearer understanding for readers.

Response 2: Thank you for your guidance and suggestions. We have detailed the limitations of existing facial recognition research in the second paragraph (lines 75 to 79) of the introduction and discussed the approaches and contributions of this study in the third paragraph (lines 93 to 101). Existing facial recognition studies are typically designed for scenarios involving a small number of students, with less attention paid to issues like pose variations and complex environmental applications. Additionally, these studies focus on student attendance management rather than using facial recognition technology to record and track student behaviors in class. The main contributions of this paper are as follows: We collected and created a classroom facial dataset and a classroom behavior dataset, and employed a pose synthesis-based data augmentation method to enhance the facial dataset. In classrooms with numerous closely arranged students, identity recognition can be confused due to angles and overlaps. To alleviate this issue, we proposed an identity recognition algorithm by analyzing the relationship between faces and bodies. We developed a system application based on embedded devices, combining facial recognition and object detection technologies, which lays the foundation for classroom analysis and personalized teaching.

Comments 3: In the section on algorithm principles, a more detailed description of the key technologies and algorithms employed is needed, including the rationale for algorithm selection, details of the model architecture, and settings of hyperparameters.

Response 3: In our system, identity verification is crucial; therefore, choosing a reliable facial recognition model was our primary task. Due to the resource-intensive nature of facial recognition models, we opted for a relatively lightweight object detection model to ensure efficient system operation. Additionally, to facilitate the integration of our solution into the Atlas 200 DK, we selected the InsightFace and YOLOv5 algorithms to work together. The choice of these two algorithms was based on their performance advantages and resource requirements to ensure stability and accuracy in performing efficient facial recognition and object detection. Details of the model architecture and the settings of hyperparameters have also been added to the manuscript, specifically in section 2.1.2.

Comments 4: The authors merely mention the algorithms used in the manuscript, but these networks lack detailed descriptions, such as the specific parameters of each layer, the manner in which layers are connected, and the input and output dimensions of the networks. It is also recommended to add an architectural diagram of the model algorithms.

Response 4: Thank you for your recommendations. We have addressed this comment by providing detailed descriptions of the networks in sections 2.1 and 2.2 of the manuscript. Additionally, architectural diagrams and equations have been added to offer a more visual representation of the algorithms. These enhancements help clarify the specific parameters of each layer, the connections between layers, and the input and output dimensions of the networks.

Comments 5: In the experimental section, the authors have not compared their work with the latest technologies or conducted performance comparisons under the same conditions. Additionally, the document does not explicitly state the specific advantages and disadvantages of the methods used in comparison to others.

Response 5: As previously indicated, one focus of our research is optimizing models for embedded devices, which led us to choose classic and mature technologies rather than extensively comparing them with the latest technologies, based on considerations for resource efficiency and deployment needs. Thank you for your suggestion. We have addressed this by adding Table 8 in Section 4.1.2, where we compare and analyze several latest technologies in terms of model performance, computational requirements, and other metrics. This allows us to clearly state the specific advantages and disadvantages of the methods used compared to others.

Comments 6: In the experimental section, the conclusion at line 529 is drawn that " and the highest false detection rate is when the number of students with their heads down is the highest. In addition to students looking down, touching and covering their faces with their hands is also a cause of false detection.", please provide a corresponding explanation for this conclusion.

Response 6: The phenomenon where students lower their heads or touch and cover their faces significantly impacts the integrity and accuracy of facial recognition, as these actions can deform or partially obscure facial features. Despite using head pose simulation enhancements to mimic different head angles and coverages in our dataset creation, the facial data under such complex circumstances in the dataset are still insufficient to fully replicate real-world scenarios. Additionally, while the physical positioning of students in a classroom setting remains relatively stable, we compensate for data limitations by continuously recording recognition results within the same area, thereby enhancing identity verification accuracy. In future work, we plan to expand our dataset to include a broader range of postures and obstructions to increase system robustness. These explanations have been added to the manuscript on page 21, lines 592-599.

Comments 7: The article conducted a system evaluation in a real classroom environment, so in the conclusion section, the paper should discuss the potential challenges that may be encountered when deploying the model in an actual classroom setting, such as real-time performance, device compatibility, and user privacy. This will help to enhance the credibility of the research and can provide guidance for future research and development efforts.

Response 7: In the final paragraph of the conclusion, we discuss a range of potential challenges that may be encountered when deploying the model in an actual classroom setting, including device compatibility, user privacy, and the diverse needs of different applications in the education sector. Additionally, we explore the possibility of adopting a multimodal approach to enhance system performance and adaptability. These discussions aim to provide direction for future research and development, emphasizing the importance of considering real-time performance and user privacy in practical applications. We hope these discussions will enhance the credibility of the research and offer valuable insights for future technological improvements and practical deployments.

Reviewer 2 Report

Comments and Suggestions for Authors

This is a very interesting study that leverages AI and big datasets to detect and trace individual student behaviors. This has significant implications for automatic detection of psychological and behavioral constructs within the classroom for personalized learning interventions. There are very few edits that are suggested for this paper. Overall, I recommend this paper be asked to do minor revisions.

·      When introducing studies, it would be good to mention the authors’ names, e.g., Smith et al. [1] instead of “literature”

·      Please explain more by what the following excerpt means: “…a key issue in this process is the over-emphasis on theoretical constructs…”

·      The authors state: “The recognition capabilities of these 58 algorithms on specific datasets have surpassed human-level performance.” How so? How has this been supported by empirical literature?

·      Not necessary, but it would be interesting to enhance the conclusion by incorporating the uses of the results of this paper to several different fields of education - for teachers, for researchers, for curriculum designers, etc.

Author Response

Comments1: When introducing studies, it would be good to mention the authors’ names, e.g., Smith et al. [1] instead of “literature”.

Response 1: Thank you for your suggestion regarding the citation format. We have carefully reviewed and revised all the references in the manuscript to ensure that each citation explicitly mentions the authors' names along with the corresponding reference numbers. This change aims to provide clearer and more accurate attributions throughout the paper.

Comments 2: Please explain more by what the following excerpt means: “…a key issue in this process is the over-emphasis on theoretical constructs…”

Response 2: We have further explained this in lines 49-55 of the introduction. The specific meaning is as follows: In the development of intelligent education, many studies and projects focus more on establishing theoretical models and building infrastructure, rather than applying these theories and facilities to educational practice. This tendency leads to a disconnect between theory and practice, hindering the full potential of intelligent technologies in actual teaching environments. This explanation aims to clarify the challenges and suggest a more balanced approach to integrating theory and practical application in the field of intelligent education.

Comments 3: The authors state: “The recognition capabilities of these 58 algorithms on specific datasets have surpassed human-level performance.” How so? How has this been supported by empirical literature?

Response 3: In 2015, Lu et al. in their paper "Surpassing Human-Level Face Verification Performance on LFW with GaussianFace" introduced a multitask learning approach based on Discriminative Gaussian Process Latent Variable Model (DGPLVM), called GaussianFace. This model achieved an accuracy of 98.52% on the LFW benchmark test, surpassing human performance on LFW, which stood at 97.53%. With the advancement of technology, the algorithms mentioned in our manuscript [6-9] have all surpassed this benchmark on the dataset, further solidifying the fact that these algorithms have exceeded human-level performance in specific datasets. This is supported by empirical literature which tracks and validates the performance improvements of these algorithms over time.

Comments 4: Not necessary, but it would be interesting to enhance the conclusion by incorporating the uses of the results of this paper to several different fields of education - for teachers, for researchers, for curriculum designers, etc.

Response 4: Thank you for your valuable suggestion. I agree that enriching the conclusion with applications of our research results across different educational fields would indeed add depth and breadth to our paper. Based on your suggestion, I have expanded the conclusion (lines 650-655) to include future prospects for educational researchers and teachers. This addition discusses how the findings can aid in curriculum development, improve teaching methodologies, and provide researchers with new tools for educational analysis. Again, thank you for your guidance.